# Molecular Landscape of Metastatic Lung Adenocarcinoma in Bulgarian Patients—A Prospective Study

**DOI:** 10.3390/ijms26147017

**Published:** 2025-07-21

**Authors:** George Dimitrov, Vladislav Nankov, Natalia Chilingirova, Zornitsa Kamburova, Savelina Popovska

**Affiliations:** 1Department of Medical Oncology, Medical University of Sofia, University Hospital “Tsaritsa Yoanna”, 1527 Sofia, Bulgaria; 2Centre of Competence in Personalized Medicine, 3D and Telemedicine, Robotic Assisted and Minimally Invasive Surgery—Leonardo da Vinci, 5800 Pleven, Bulgaria; vladislav.nankov@mu-pleven.bg (V.N.); zornitca.kamburova@mu-pleven.bg (Z.K.); savelina.popovska@mu-pleven.bg (S.P.); 3Department of Anatomy, Histology, Cytology and Biology, Medical University of Pleven, 5800 Pleven, Bulgaria; 4Medical Oncology Clinic, Heart and Brain Hospital, 5804 Pleven, Bulgaria; n.chilingirova.pn@heartandbrain.bg; 5Department of Medical Genetics, Medical University of Pleven, 5800 Pleven, Bulgaria; 6Department of Clinical Pathology, Medical University of Pleven, 5800 Pleven, Bulgaria

**Keywords:** lung adenocarcinoma, molecular profiling, driver mutations, Bulgarian population

## Abstract

Lung adenocarcinoma exhibits a heterogeneous molecular landscape shaped by key oncogenic drivers and tumor suppressor gene alterations. Mutation frequencies vary geographically, influenced by genetic ancestry and environmental factors. However, the molecular profile of lung adenocarcinoma in Bulgarian patients remains largely uncharacterized. We conducted a prospective study of 147 Bulgarian patients with metastatic lung adenocarcinoma, analyzing clinicopathologic features and somatic mutation frequencies using next-generation sequencing. Key mutations and their prevalence were assessed and compared with published data from other populations. The cohort included predominantly male patients (68.0%) with a median age of 67 years. *TP53* mutations were most frequent (41.5%), followed by *EGFR* alterations (19.0%) and *KRAS* c.34G>T (p.Gly12Cys) (17.0%). Over half of the patients (51.0%) harbored two or more gene mutations. Mutation frequencies aligned closely with European cohorts, exhibiting a lower prevalence of *EGFR* mutations compared to East Asian populations. This study characterizes the molecular landscape of lung adenocarcinoma in Bulgaria, highlighting the predominance of *TP53* and *KRAS* mutations. The findings emphasize the need for comprehensive molecular profiling to inform targeted therapies and support precision oncology approaches tailored to the Bulgarian population. Further research is needed to validate these results and improve clinical outcomes.

## 1. Introduction

Lung adenocarcinoma is characterized by a highly heterogeneous molecular landscape, shaped by recurrent alterations in key oncogenic drivers and tumor suppressor genes. The most frequently mutated genes include *EGFR*, *KRAS*, *ALK*, *BRAF*, *ERBB2* (HER2), *MET*, *ROS1*, and *RET*, with *EGFR* and *KRAS* mutations being the most prevalent. However, their frequency varies substantially across populations, influenced by genetic ancestry, smoking status, and environmental exposures. Additional alterations of clinical and biological relevance involve *TP53*, *NF1*, *RBM10*, *ARID1A*, and *STK11*, along with established oncogenic events such as *ALK*, *ROS1*, and *RET* fusions and *MET* exon 14 skipping mutations [1,2].

The molecular evolution of lung adenocarcinoma generally follows a stepwise model. Early and preinvasive lesions—including atypical adenomatous hyperplasia, adenocarcinoma in situ, and minimally invasive adenocarcinoma—typically harbor clonal mutations in canonical driver genes such as *EGFR*, *ERBB2*, *NRAS*, and *BRAF*. In contrast, alterations in *TP53* and genes associated with cellular motility and metastatic potential tend to arise later, contributing to subclonal diversification and disease progression. This evolution is accompanied by increasing genomic instability, including accumulation of copy number alterations and global DNA hypomethylation [3,4,5].

Transcriptomic profiling has further refined lung adenocarcinoma classification, delineating molecular subtypes with distinct patterns of pathway activation, immune microenvironment composition, and clinical outcomes. For instance, subtypes enriched for *EGFR* mutations are typically associated with a favorable prognosis, whereas those defined by *TP53* mutations and high genomic instability exhibit more aggressive clinical behavior and are often resistant to immunotherapy [6,7].

Importantly, the prevalence and distribution of these alterations vary considerably across geographic regions and ethnic groups. In East Asian populations, *EGFR* mutations are detected in up to 55% of lung adenocarcinomas, while *KRAS* mutations are less frequent. Alterations in *ALK*, *RET*, and *ERBB2* are also more common, particularly among never-smokers. East Asian tumors tend to exhibit greater genomic stability and, in some cases, display distinct immune-related transcriptomic profiles [8,9].

By contrast, European and North American cohorts show a higher prevalence of *KRAS* mutations—especially the G12C variant—and lower frequencies of *EGFR* alterations. These populations typically have a higher overall tumor mutational burden, particularly among smokers, and a greater prevalence of *STK11* mutations. The broader spectrum of actionable alterations in these groups highlights the need for comprehensive molecular diagnostics [10,11].

Latin American populations demonstrate additional distinct patterns, including higher frequencies of somatic variants such as *SLC36A4*, *AP1S1*, and *TP53*, as well as a greater burden of germline cancer susceptibility variants. In Mexican patients, for example, the classical smoking-associated mutational signature (SBS4) is less prominent, even among smokers, suggesting alternative carcinogenic mechanisms. Within-country heterogeneity has also been observed; for instance, in China, *EGFR* mutations are more common in eastern regions, while *KRAS* mutations predominate in the south [12]. These global and regional differences underscore the importance of localized molecular profiling to inform therapeutic decision-making and optimize outcomes.

In Bulgaria, however, the molecular landscape of lung adenocarcinoma remains insufficiently characterized. To date, no large-scale genomic profiling studies have focused specifically on Bulgarian patients. Consequently, assumptions regarding the mutation spectrum in Bulgaria are often extrapolated from broader European data [10,13,14]. To address this knowledge gap, we conducted a prospective analysis of the mutation landscape in Bulgarian patients with lung adenocarcinoma. Our aim was to define the regional molecular profile and explore its implications for targeted therapy and precision oncology in this understudied population.

## 2. Results

Between January 2020 and April 2025, a total of 216 patients with metastatic NSCLC underwent comprehensive molecular profiling. Patients were initially assessed for histological subtype and availability of adequate tumor material. A total of 69 patients were excluded from the analysis: 21 patients due to insufficient tumor material for molecular testing, 6 patients due to inconclusive molecular results, 4 patients with confirmed small cell lung carcinoma (SCLC), and 38 patients with squamous cell carcinoma (SCC) histology. After applying these exclusion criteria, 147 patients with histologically confirmed, newly diagnosed, and treatment naïve metastatic lung adenocarcinoma were included. Complete molecular profiling data were included in the final cohort for analysis (Figure 1).

Table 1 summarizes the clinical and genomic characteristics of the 147 patients included in the study. The cohort consisted predominantly of males (68.0%) with a median age of 67 years (±9.16 SD). Regarding smoking history, 41.5% were current or former smokers, while 58.5% reported never smoking. Tissue samples were obtained predominantly from primary lung tumor biopsies (73.4%). Tumor grading showed that the majority of tumors were high grade (G3) at 71.4%, with the remaining 28.6% classified as lower grade (<G3). Genomic profiling revealed diverse mutation frequencies: *TP53* was the most frequently mutated gene, detected in 41.5% of cases (n = 61). A wide spectrum of mutations was observed, including missense (52.5%), nonsense (26.2%), frameshift (13.1%), and splice site variants (8.2%), predominantly affecting the DNA-binding domain of the protein. Recurrent hotspot mutations included p.Arg273 (observed as p.Arg273His, p.Arg273Leu, p.Arg273Ser, and p.Arg273Cys, n = 5), p.Val157Phe (n = 3), p.Cys242Phe (n = 2), p.Gly266Val or Ter (n = 3), and p.Gly154 (p.Gly154Val or frameshift, n = 3). Other pathogenic variants included p.Arg282Trp, p.Arg175His, p.Glu258Lys, p.Arg249Ser, p.Gly245Cys, and p.Arg213Ter. Several splice-disrupting alterations were detected, including c.673-1G>T, c.673-1G>C, c.376-1G>A, and c.920-1G>T, which are predicted to affect mRNA processing. Two patients harbored the synonymous variant p.Thr125, and a subset showed compound alterations. The *KRAS* mutation c.34 G>T (p.Gly12Cys) was found in 17.0% (n = 25) of patients. *EGFR* alterations (total n = 28) were represented primarily by exon 19 deletions and *MET* mutations, each present in 8.1% (n = 12) of the cohort. Other notable mutations included *PIK3CA*, *STK11*, and *CDKN2A*, each detected in approximately 7.4% of patients (n = 11), followed by *EGFR* exon 21 L858R mutations at 6.1% (n = 9), and *BRAF* mutations in 4.7% (n = 7). Less frequent mutations (below 3%) included *JAK3* (4.1%), *ATM* (3.4%), *KDR* (3.4%), and multiple others with individual mutation frequencies below 2%. Notably, over half of the patients (51.0%, n = 75) harbored two or more gene mutations (Figure 2).

Figure 3 illustrates the distribution of somatic mutations in key oncogenic drivers and tumor suppressor genes among male (n = 100) and female (n = 47) patients with metastatic lung adenocarcinoma. Notably, *EGFR* mutations were more frequently observed in female patients, particularly *EGFR* exon 19 deletions (19.1% in females vs. 3% in males) and EGFR exon 18 *E709K* (2.1% vs. 0%), consistent with prior reports linking EGFR alterations to female sex. In contrast, *KRAS* c.34G>T (p.Gly12Cys) mutations were significantly more common in males (17% vs. 8.5%). Mutations in *TP53* were the most prevalent overall and occurred more frequently in males (43%) than females (38.3%), while *PIK3CA*, *STK11*, *MET*, and *CDKN2A* mutations were also more commonly detected in the male subgroup. Interestingly, *NOTCH1*, *MAP2K1*, and *PIK3R1* alterations were detected exclusively in females, albeit at low frequency.

Figure 4 compares the distribution of somatic mutations in major oncogenic and tumor suppressor genes between smokers (n = 61) and non-smokers (n = 86) with metastatic lung adenocarcinoma. A marked divergence in the mutational landscape was observed between the two groups. *EGFR* mutations were significantly more prevalent in non-smokers. The most common alterations included EGFR exon 19 deletions (14.0% in non-smokers vs. 0% in smokers), *EGFR* exon 21 L858R (9.3% vs. 1.6%), and *EGFR* exon 18 E709K (1.2% vs. 0%). These findings are consistent with established evidence linking EGFR-driven oncogenesis to never-smoker status. Conversely, *KRAS* c.34G>T (p.Gly12Cys) mutations were strongly associated with smoking history, detected in 39.3% of smokers compared to only 1.2% of non-smokers, confirming the known relationship between tobacco exposure and RAS-pathway activation. Mutations in *STK11*, *TP53*, and *CDKN2A* were also more frequent in smokers, reflecting the higher mutational burden typically observed in tobacco-related lung cancers. Non-smokers showed relatively enriched frequencies of *PIK3CA*, *MET*, *RET*, and *NOTCH1* mutations, suggesting alternative oncogenic drivers in the absence of smoking-related carcinogens. Co-mutations in *PIK3R1*, *MAP2K1*, and *VHL* were exclusively found in non-smokers, albeit at low prevalence.

Figure 5 illustrates the distribution of somatic mutations in metastatic lung adenocarcinoma stratified by histological tumor grade—Grade 3 (poorly differentiated; n = 105) versus Grades 1–2 (well to moderately differentiated; n = 42). Distinct molecular patterns were observed across tumor grades, reflecting underlying biological differences. *TP53* mutations were markedly more prevalent in Grade 3 tumors, observed in 42.9% of cases compared to 38.1% in Grades 1–2, supporting the role of *TP53* in high-grade tumor progression and genomic instability. Similarly, *KRAS* p.Gly12Cys mutations were more frequent in Grade 3 tumors (15.2%) versus 7.6% in lower-grade tumors, in line with its known association with aggressive tumor behavior. Grade 3 tumors also harbored higher frequencies of *STK11*, *CDKN2A*, *ATM*, *EZH2*, *NOTCH1*, and *PIK3CA* mutations, suggesting a more complex genomic landscape associated with dedifferentiation and potentially worse prognosis. Co-occurring alterations in genes such as *PIK3R1*, *MAP2K1*, and *SMAD4* were almost exclusively identified in high-grade tumors. In contrast, *EGFR* mutations (including exon 19 deletions and L858R substitutions) were found at comparable rates across tumor grades but showed a slight enrichment in Grades 1–2, consistent with the observation that *EGFR*-mutated adenocarcinomas often present with lepidic or acinar patterns typical of well to moderately differentiated tumors.

## 3. Discussion

This prospective study provides the first comprehensive characterization of the molecular landscape of metastatic lung adenocarcinoma in a Bulgarian cohort, revealing both expected and distinctive features that are consistent with—and in some aspects extend—published data from other populations. Our findings reinforce the high molecular heterogeneity of lung adenocarcinoma, underscoring its biologic complexity and the wide spectrum of oncogenic driver and tumor suppressor gene alterations that influence disease behavior and therapeutic response. Consistent with global reports, *TP53* mutations were the most frequent aberration, detected in 41.5% of cases, highlighting its central role in tumorigenesis, genomic instability, and disease progression [15]. *KRAS* mutations, particularly the G12C variant (17%), were also prevalent and aligned with the frequencies reported in European and North American cohorts, where *KRAS* alterations tend to predominate over *EGFR* mutations [16]. In contrast, EGFR alterations—across all types—were detected in only 19% of patients in our cohort, a frequency substantially lower than that observed in East Asian populations, where *EGFR* mutation rates commonly range between 40% and 55% [17]. This disparity underscores the critical role of ethnic and geographic variability in shaping mutation prevalence, likely influenced by genetic ancestry, lifestyle, and environmental exposures such as tobacco use, affecting 41.5% of our cohort. Mutations in other clinically actionable genes, including *MET*, *PIK3CA*, *STK11*, and *CDKN2A*, were observed at rates comparable to those in other European datasets [18,19,20]. These findings further support the concept of oncogene addiction in NSCLC—a phenomenon where tumor cells remain highly dependent on specific driver mutations for survival—and highlight the clinical importance of identifying these dependencies with high-resolution molecular tools [21]. Notably, more than half of the patients in our cohort (51%) harbored two or more genomic alterations, highlighting the frequent occurrence of co-mutations and underscoring the substantial intertumoral heterogeneity that characterizes lung adenocarcinoma. This genomic complexity has critical implications in the era of precision oncology, as co-occurring alterations can modulate tumor biology, influence sensitivity to targeted therapies, and contribute to both primary and acquired resistance mechanisms. Understanding these mutational interactions is increasingly important for optimizing treatment sequencing, anticipating resistance pathways, and guiding the design of combinatorial therapeutic strategies tailored to the molecular profile of individual tumors [22,23].

The clinical implications of these results are significant. The predominance of *KRAS* and *TP53* mutations, coupled with the lower frequency of targetable *EGFR* alterations, suggests that targeted therapy strategies in the Bulgarian population should prioritize emerging KRAS G12C inhibitors and rational combination approaches targeting TP53-driven pathways [24]. Furthermore, although less common, actionable mutations in *MET*, *PIK3CA*, and *ALK* fusion events offer opportunities for enrollment in biomarker-selected trials and the use of approved targeted agents. Moreover, our cohort’s molecular profile shares greater resemblance with Western than East Asian populations, reinforcing the need for region-specific data to avoid reliance on extrapolated global frequencies that may not reflect local realities. Population-specific molecular epidemiology is indispensable to ensuring that testing algorithms, treatment guidelines, and health policy are aligned with the actual mutational burden and therapeutic opportunities within a given demographic [25,26,27,28]. Importantly, this study underscores the critical need for broad, comprehensive NGS over limited hotspot panels or single-gene assays. Given the substantial inter-individual and inter-population variability in actionable genomic alterations, a wide-ranging molecular approach is essential for the accurate identification of oncogene-addicted tumors and for guiding evidence-based, personalized therapeutic decisions. Narrow testing strategies risk missing uncommon but targetable mutations—such as those in *KRAS*, *HER2*, *RET*, *MET*, and *NTRK*—that may profoundly impact clinical management and outcomes. As the repertoire of targeted therapies continues to expand, comprehensive genomic profiling ensures that patients are not excluded from potentially life-extending treatments due to incomplete molecular characterization. Moreover, from a health systems perspective, broader NGS may ultimately prove more cost-effective by enabling more precise therapy selection, avoiding ineffective treatments, and supporting equitable access to precision oncology [29].

In selecting the Archer^®^ VariantPlex^®^ Somatic Assay, we prioritized a method that balances clinical relevance, cost-effectiveness, and analytical performance. This assay involves working with simple lyophilized reagents during library preparation, which reduces the risk of contamination and eliminates the need for master mixes. The panel utilizes Anchored Multiplex PCR (AMPTM) technology for reliable DNA sequencing. AMP’s special features ensure enhanced performance of tasks like error correction, molecular counting, and detection of low allele frequency mutations. Unlike traditional priming methods, AMP chemistry enables primer hybridization to both strands of DNA, providing optimal coverage of targeted regions for amplification and characterization of challenging alterations. AMP chemistry also utilizes molecular barcode adapters that bind to DNA fragments before amplification. That enables efficient amplification of target sequences from degraded or fragmented DNA from FFPE tissue [30].

The primary goal of our study was to identify clinically relevant mutations. While it has been previously reported that large gene panels (in the range of hundreds of genes) can be used to detect more variants compared with medium-sized and small hotspot panels (i.e., up to 50 genes or similar), the additional variants often have no direct impact on patient management. Thus, the Archer^®^ VariantPlex^®^ Somatic Assay provides an efficient and cost-effective alternative for the detection of clinically relevant mutations in cancer. Larger panels are more time-consuming, and they often involve a more complex data analysis workflow. Smaller gene panels, like the Archer^®^ VariantPlex^®^ Somatic Assay, have a shorter turnaround time as they are less “sequencing intensive”, and data analysis is also less complex. Targeted panel sequencing can allow for deeper sequencing (higher number of reads covering the same region of the genome) than WGS (whole-genome sequencing) and WES (whole-exome sequencing). This focused approach allows for higher sensitivity for specific mutations and ensures the detection of even rare variants [31].

Among the key strengths of this study are its prospective design, the use of a clinically validated broad targeted NGS panel, and its contribution of foundational data on the genomic landscape of advanced lung adenocarcinoma in a Bulgarian population, a demographic for which molecular data have been notably scarce. The prospective nature of the study allowed for standardized sample collection, uniform molecular testing, and minimized retrospective bias. The application of a comprehensive NGS assay enabled the detection of a wide spectrum of actionable and co-occurring mutations, providing insights into both common and rare oncogenic drivers. Importantly, by profiling a nationally representative patient cohort, this study offers the first detailed molecular snapshot of metastatic lung adenocarcinoma in Bulgaria, thereby filling a critical knowledge gap in Eastern and Southeastern European cancer genomics. These attributes collectively enhance the translational relevance and generalizability of our findings, supporting the development of regionally informed precision oncology strategies and facilitating future inclusion of this population in global clinical research efforts.

Nonetheless, this study is not without limitations. The relatively modest sample size and the absence of clinical outcome data, such as survival or treatment response, limit the ability to draw definitive prognostic or predictive conclusions. These findings should therefore be validated in larger, multi-center prospective cohorts with longitudinal follow-up to fully assess their clinical utility. Another important limitation involves the technical approach used for DNA sequencing. Specifically, this study employed an amplicon-based library preparation method, which is susceptible to certain artifacts, most notably, allele dropout. This phenomenon occurs when base pair mismatches or small insertions/deletions (indels) are present in primer binding sites, thereby impairing primer annealing. As a result, these amplicons may fail to amplify efficiently, leading to reduced sequencing coverage and inaccurate variant allele frequency estimates. Additionally, regions of high guanin–cytosine (GC) content or highly repetitive sequences pose challenges for amplification, further contributing to data loss or bias. Indels that remove primer-binding sites may be completely missed, and low-quality reads at amplicon termini can lead to miscalling of variants.

To address the inherent limitations of amplicon-based sequencing, hybrid capture-based enrichment methods present a more robust alternative. These approaches employ longer, sequence-specific capture probes that hybridize to target genomic regions with higher tolerance for mismatches, such as single-nucleotide polymorphisms or small insertions and deletions within primer-binding sites. As a result, they significantly reduce the risk of allele dropout—a common challenge in amplicon-based assays—and enhance the accuracy of variant detection. Furthermore, hybrid capture enables more uniform coverage across GC-rich or repetitive regions and supports the reliable identification of a wider spectrum of variant types, including larger insertions, deletions, and structural rearrangements. This makes hybrid capture particularly well-suited for comprehensive genomic profiling in clinical oncology settings, especially when working with heterogeneous or degraded samples, such as those derived from FFPE tissue [32]. Future studies may benefit from integrating such techniques to enhance analytical accuracy, particularly when comprehensive and unbiased variant detection is critical for clinical decision-making.

## 4. Materials and Methods

### 4.1. Patient Selection

This prospective study enrolled patients with histologically confirmed newly diagnosed metastatic non-small cell lung cancer (NSCLC) who were referred for comprehensive molecular profiling at the Centre of Competence in Personalized Medicine, 3D and Telemedicine, Robotic-Assisted and Minimally Invasive Surgery—Leonardo da Vinci in Pleven, Bulgaria, between January 2020 and April 2025. Eligible participants were adults (aged ≥ 18 years) with a pathological diagnosis of lung adenocarcinoma and radiologically confirmed stage IV disease at the time of initial presentation. Histopathological diagnosis was established based on standard morphological criteria, including evidence of glandular differentiation, mucin production, and characteristic architectural patterns such as lepidic, acinar, papillary, micropapillary, and solid growth. In cases displaying a solid pattern, where differential diagnosis with squamous cell carcinoma was necessary, immunohistochemistry was performed using an antibody against thyroid transcription factor-1 (TTF-1; clone 8G7G3/1, DAKO, Glostrup, Denmark) to support adenocarcinoma lineage. Only tumors exhibiting adenomatous or solid growth patterns and demonstrating positive nuclear immunoreactivity for TTF-1 were included, ensuring diagnostic consistency with primary pulmonary adenocarcinoma. Further inclusion criteria required the availability of formalin-fixed, paraffin-embedded (FFPE) tumor tissue specimens with adequate tumor content (≥10% viable tumor cells), meeting predefined quality control standards for nucleic acid extraction and next-generation sequencing. All tissue samples were centrally reviewed by an experienced thoracic pathologist prior to molecular analysis to ensure uniformity in diagnostic criteria and sample adequacy. Patients were excluded if they had non-adenocarcinoma histology (i.e., small cell lung carcinoma or squamous cell carcinoma), if tumor material was insufficient or failed quality control for next-generation sequencing, or if complete clinical or pathological data were unavailable. A total of 216 patients with NSCLC were screened for eligibility.

### 4.2. Genomic Analysis and Molecular Profiling

Genomic DNA was extracted from formalin-fixed, paraffin-embedded (FFPE) tumor specimens using the QIAamp^®^ DNA FFPE Advanced Kit (Qiagen, Hilden, Germany), following the manufacturer’s protocol. DNA concentration and purity were assessed using the Qubit™ dsDNA High Sensitivity Assay Kit (Thermo Fisher Scientific, Waltham, MA, USA) and the Archer PreSeq™ DNA Quality Control Assay (Integrated DNA Technologies, Coralville, IA, USA). All DNA samples were stored at −80 °C until library preparation and sequencing. Targeted next-generation sequencing (NGS) was performed to detect single-nucleotide variants (SNVs), insertions and deletions (indels), and copy number variations (CNVs) across a panel of 67 cancer-related genes using the Archer^®^ VariantPlex^®^ Somatic Assay for Illumina (Integrated DNA Technologies, Coralville, IA, USA). For each sample, 360 ng of genomic DNA was used for library preparation. This process included enzymatic DNA fragmentation, end repair, A-tailing, and ligation of platform-specific molecular barcoded adapters, performed according to the manufacturer’s standardized protocol. Target enrichment was accomplished using Anchored Multiplex PCR (AMP, Louisville, CO, USA) technology. The final pooled libraries were loaded at a concentration of 15 pM and sequenced on the MiSeqDx platform (Illumina, San Diego, CA, USA). Data analysis, including somatic variant calling and functional annotation, was conducted using Archer Analysis version 6.2.7 (Integrated DNA Technologies, Coralville, IA, USA). To ensure high-confidence mutation detection, variant filtering criteria included a minimum allelic frequency threshold of ≥5% and a minimum variant read depth of ≥500×.

### 4.3. Statistical Analysis

Descriptive statistics were employed to characterize the genomic landscape. Categorical variables are presented as frequencies and percentages, while continuous variables are summarized using median and range.

### 4.4. Ethical Considerations

This study was approved by the Institutional Ethics Board of the Medical University of Pleven and conducted in accordance with the ethical principles outlined in the Declaration of Helsinki. Written informed consent was obtained from all patients before participation, and data were anonymized to protect patient confidentiality.

## 5. Conclusions

This prospective study provides the first detailed insight into the molecular landscape of lung adenocarcinoma in Bulgarian patients, revealing a mutation profile largely consistent with European cohorts but distinct from East Asian populations. The predominance of *TP53* and *KRAS* mutations, alongside a lower frequency of *EGFR* alterations, underscores the importance of population-specific genomic profiling. Our findings support the implementation of comprehensive molecular diagnostics in Bulgaria to guide precision oncology and improve targeted treatment strategies tailored to the unique genetic and environmental context of this population. Further large-scale studies are warranted to validate these results and assess their clinical impact.

## Figures and Tables

**Figure 1 ijms-26-07017-f001:**
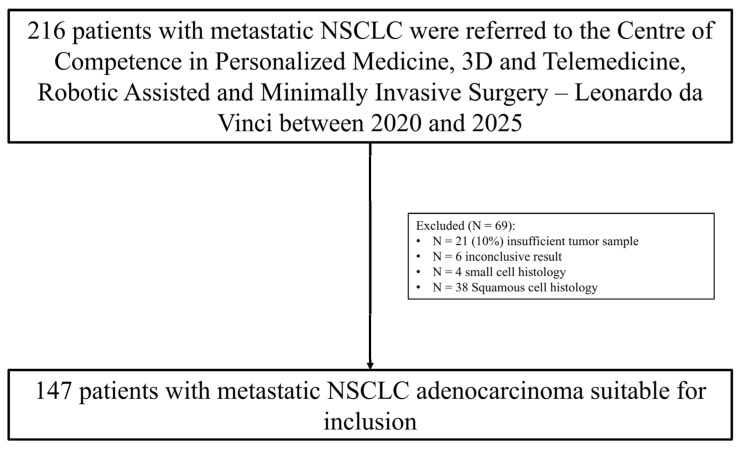
Overview of patient selection process.

**Figure 2 ijms-26-07017-f002:**
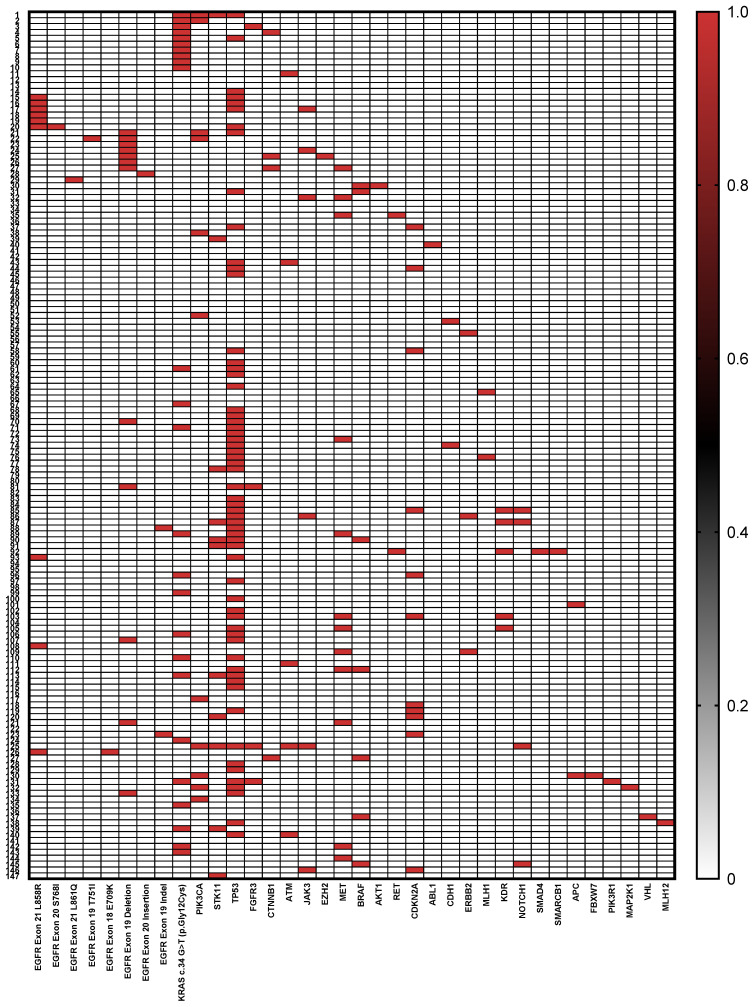
Heat-map of mutation landscape in the 147 Bulgarian patients with stage IV lung adenocarcinoma.

**Figure 3 ijms-26-07017-f003:**
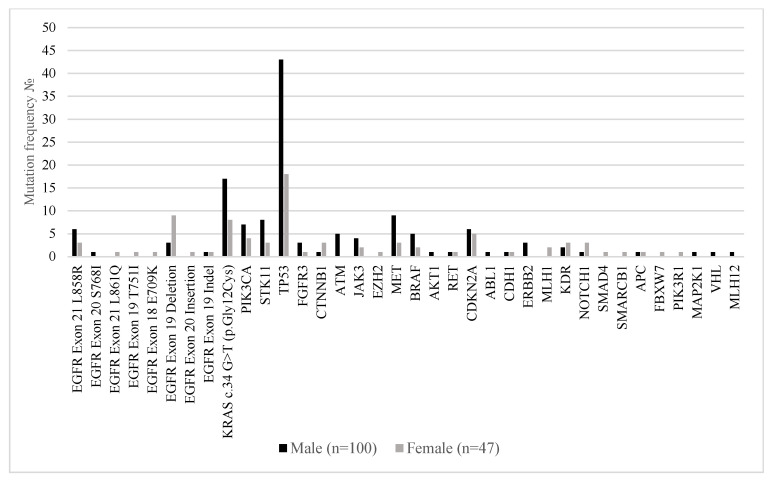
Mutation frequencies in the 147 Bulgarian patients with stage IV lung adenocarcinoma by sex.

**Figure 4 ijms-26-07017-f004:**
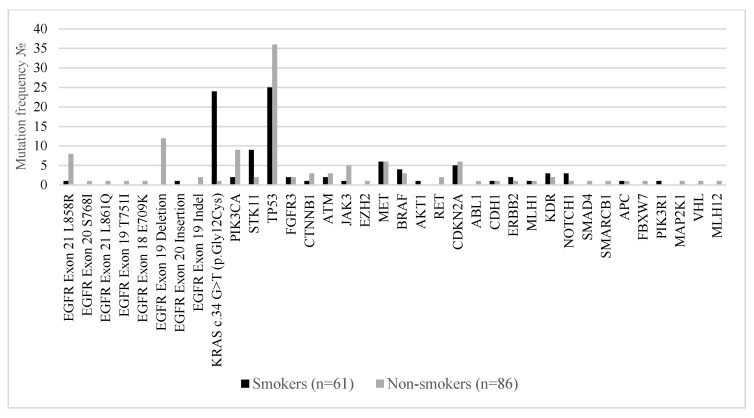
Mutation frequencies in the 147 Bulgarian patients with stage IV lung adenocarcinoma by smoking status.

**Figure 5 ijms-26-07017-f005:**
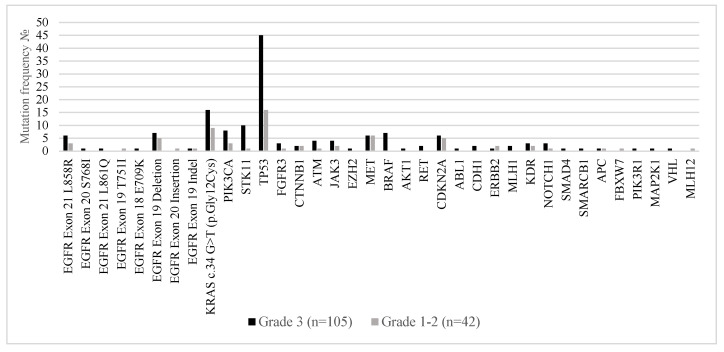
Mutation frequencies in the 147 Bulgarian patients with stage IV lung adenocarcinoma by tumor grade.

**Table 1 ijms-26-07017-t001:** Patient characteristics (n = 147).

Characteristic	n (%)
Sex	
Female	47 (32.0%)
Male	100 (68.0%)
Median age, years	67 (±9.16 SD)
Smoking history	
Current or past	61 (41.5%)
Never-smoker	86 (58.5%)
Tumor grade	
<G3	42 (28.6%)
G3	105 (71.4%)
Mutation frequencies	
TP53	61 (41.5%)
KRAS c.34 G>T (p.Gly12Cys)	25 (17.0%)
EGFR Exon 19 Deletion	12 (8.1%)
MET	12 (8.1%)
PIK3CA	11 (7.4%)
STK11	11 (7.4%)
CDKN2A	11 (7.4%)
EGFR Exon 21 L858R	9 (6.1%)
BRAF	7 (4.7%)
JAK3	6 (4.1%)
ATM	5 (3.4%)
KDR	5 (3.4%)
FGFR3	4 (2.7%)
CTNNB1	4 (2.7%)
NOTCH1	4 (2.7%)
ERBB2	3 (2.0%)
EGFR Exon 19 Indel	2 (1.3%)
RET	2 (1.3%)
CDH1	2 (1.3%)
MLH1	2 (1.3%)
APC	2 (1.3%)
EGFR Exon 20 S768I	1 (0.7%)
EGFR Exon 21 L861Q	1 (0.7%)
EGFR Exon 19 T751I	1 (0.7%)
EGFR Exon 18 E709K	1 (0.7%)
EGFR Exon 20 Insertion	1 (0.7%)
EZH2	1 (0.7%)
AKT1	1 (0.7%)
ABL1	1 (0.7%)
SMAD4	1 (0.7%)
SMARCB1	1 (0.7%)
FBXW7	1 (0.7%)
PIK3R1	1 (0.7%)
MAP2K1	1 (0.7%)
VHL	1 (0.7%)
MLH12	1 (0.7%)
≥2 gene mutations	75 (51.0%)

## Data Availability

The raw dataset presented in this article is available upon request.

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
