# Peer review of "Molecular Landscape of Metastatic Lung Adenocarcinoma in Bulgarian Patients—A Prospective Study"

_ijms, 2025, doi:10.3390/ijms26147017_

Round 1

Reviewer 1 Report

Comments and Suggestions for Authors

Methods:

  1. The authors should clearly state  that the investigated group consists  of patients with metastatic lung adenocarcinoma a subset of NSCLC, rather than implying inclusion of all NSCLC subtypes.
  2. The authors should clarify whether the group consisted of newly diagnosed, before treatment patients or whether some patients had received  therapy before molecular profiling.
  3. The authors should explained whethere the samples analyzed were obtained from primary tumors, metastatic sites, or a combination of both.
  4. Ethical considerations are not addressed; the authors should state whether the study was approved by an ethics committee and whether informed consent was obtained from all participants. ( In the section results first sentence is about period of time, but it should be placed rather in description of "patient selection" or in paragraph   "ethical consideration")

Results: 

  1. The  description of patient characteristics and mutation frequencies largely duplicates the information already presented in Table 1. The authors should focus on emphasizing the most clinically significant findings.

  2. Descriptive statistics should be presented in the paper to support interpretation and discussion, while full data details could stay in the tables and figures.

  3.  In section results add interpretation.

Discussion is very good part of this paper.

Author Response

Reviewer 1:

  1. The authors should clearly state that the investigated group consists of patients with metastatic lung adenocarcinoma a subset of NSCLC, rather than implying inclusion of all NSCLC subtypes.

We thank the reviewer for this comment. We further clarified that the analysis included only metastatic adenocarcinoma NSCLC patients. Figure 1 has been revised as well.

  1. The authors should clarify whether the group consisted of newly diagnosed, before treatment patients or whether some patients had received therapy before molecular profiling.

This has been clarified in the revised Methods section. All patients included were newly diagnosed and had not received systemic anticancer therapy prior to molecular profiling.

  1. The authors should explained whethere the samples analyzed were obtained from primary tumors, metastatic sites, or a combination of both.

We have added a sentence specifying that the analyzed samples were obtained predominantly from primary lung tumor biopsies (n=108/147; 73.4%). This information is now included in the revised manuscript.

  1. Ethical considerations are not addressed; the authors should state whether the study was approved by an ethics committee and whether informed consent was obtained from all participants. (In the section results first sentence is about period of time, but it should be placed rather in description of "patient selection" or in paragraph "ethical consideration").

A dedicated Ethics Statement has been added under the “Materials and Methods” section, indicating institutional ethics committee approval and written informed consent from all participants. The suppl. material provided with the initial submission is the consent form the patients had to sign prior to study inclusion.

  1. The description of patient characteristics and mutation frequencies largely duplicates the information already presented in Table 1. The authors should focus on emphasizing the most clinically significant findings.

We appreciate the reviewer’s perspective on this point. The narrative accompanying Table 1 was made more concise to orient the reader by summarizing key demographic and main genomic features of the overall population (n=147). However, subsequent descriptions in the Results section are not redundant but instead highlight findings from focused subgroup analyses (by sex, smoking status, and tumor grade), each of which uncovers distinct patterns in mutation distribution that may have clinical implications. That said, we are committed to improving the manuscript and would be grateful if the reviewer could kindly specify which parts they believe could be refined or omitted. We will be more than happy to revise accordingly in the next version.

  1. Descriptive statistics should be presented in the paper to support interpretation and discussion, while full data details could stay in the tables and figures.

Descriptive statistics (such as frequencies and percentages) are indeed included in the respective descriptions of tables and figures throughout the Results section to contextualize key findings and support the discussion. To ensure clarity and relevance, we have focused on presenting the most salient statistics directly in the text, while detailed breakdowns remain in tabular and graphical form. If the reviewer had a particular variable or section in mind where additional descriptive statistics would enhance interpretation, we would be grateful for further clarification and will gladly incorporate the suggested additional changes.  

Discussion is very good part of this paper.

Thank you!

Reviewer 2 Report

Comments and Suggestions for Authors

The article by G. Dimitrov et al. "Molecular Landscape of Metastatic Lung Adenocarcinoma in Bulgarian Patients – A Prospective Study" should be of interest to specialists in medical genetics, oncogenetics and representatives of pharmaceutical companies developing and producing drugs for targeted cancer therapy. Studies such as the presented work are of great socio-economic importance, participating in the formation of a national health care system through targeted training of general practitioners and diagnosticians. The prevalence of a certain spectrum of mutations in a population determines the choice of diagnostic and prognostic test systems and influences national policy in the field of clinical recomendations, the choice of therapeutic and surgical strategies for patient management. In the public domain there are mainly data on the frequency of mutations in oncological diseases in the populations of Western Europe and the UK. Similar data on the countries of Southern and Eastern Europe are practically absent in the English-language literature, which makes the work of G. Dimitrov et al. particularly relevant. Despite the high value of the presented article, there are several comments.

There are several commercially available targeted NGS panels for lung cancer: the lung cancer compact panel™ (LCCP, DNA Chip Research Inc., Tokyo), FoundationOne® CDx, FFPE OncoScan, Parseq, Amoy, Oncomine Focus Assay, MSK-IMPACT (Memorial Sloan Kettering Cancer Center). Why did the authors choose the Archer® Vari-133 antPlex® Somatic Assay? Why was exome sequencing not performed? Please include the answers to these questions in the Discussion part. Additionally, I would like to know whether the authors used only the data generated by automatic processing for clinical practice, or did they analyze the raw data? Sometimes working with raw fasta (fastq) sequencing data provides additional information on mutations.

Did the work analyze only somatic mutations that arose in the tumor, or were allelic variants in tumor-associated genes, typical for the Bulgarian population, assessed? Is there information on the frequency of allelic variants in the population?

I would ask the authors to include in the discussion of the results the summary data on all mutations in the EGFR gene, regardless of the type of mutation and exon. It is possible that the total contribution of mutations in EGFR will be comparable to the contribution of mutations in TP53. What mutations were found in TP53?

Figure 2. Please increase the font size in the names of genes (mutations).

Figures 3-5. The names of genes (mutations) are given in gray. Please change the color of the captions to black. And I would ask the authors to pay attention that all mutations in the EGFR gene in these figures are designated as "EGFR...". This makes it impossible to compare the figure with the text of the article. Please correct it. In addition, it seems to me that there is no caption for the KRAS gene in all three figures.

I also would ask that the figures be placed after their first mention in the text. This will make it easier for the reader to perceive the information.

Author Response

Reviewer 2:

  1. There are several commercially available targeted NGS panels for lung cancer: the lung cancer compact panel™ (LCCP, DNA Chip Research Inc., Tokyo), FoundationOne® CDx, FFPE OncoScan, Parseq, Amoy, Oncomine Focus Assay, MSK-IMPACT (Memorial Sloan Kettering Cancer Center). Why did the authors choose the Archer® Vari-133 antPlex® Somatic Assay? Why was exome sequencing not performed? Please include the answers to these questions in the Discussion part. Additionally, I would like to know whether the authors used only the data generated by automatic processing for clinical practice, or did they analyze the raw data? Sometimes working with raw fasta (fastq) sequencing data provides additional information on mutations.

Sequencing data generated by the The Archer® VariantPlex® Somatic Assay is processed using Archer Analysis Software: a complete bioinformatics platform that can detect unique sequence fragments, enabling error correction, read deduplication, high-confidence alignment and mutation calling. Archer Analysis requires demultiplexed FASTQ files obtained straight from the sequencer as input. It utilizes unique molecular barcode adapters for duplicate read binning, error correction and read deduplication. The analysis reports contain information about sequencing metrics and number of unique observations that support each called variant.

There is currently no data about the role and prevalence of somatic mutations in tumour-associated genes in the Bulgarian population. This study has used a medium-sized gene panel (67 genes) to detect specific gene alterations that are linked to sustained proliferative signalling, evasion of apoptosis, invasion and metastasis of tumour cells. To obtain more detailed data about the occurrence of mutations in other tumour-associated genes, larger gene panels that provide comprehensive genomic profiling are needed. Our laboratory has recently completed competence testing for TruSightTM Oncology Comprehensive (EU), which is a CE-marked IVD kitted solution for comprehensive genomic profiling in oncology. We are currently looking to utilise this panel to obtain coverage of multiple variant classes in coding regions of cancer-related genes and genomic signatures in Bulgarian patients with NSCLC.

  1. Did the work analyze only somatic mutations that arose in the tumor, or were allelic variants in tumor-associated genes, typical for the Bulgarian population, assessed? Is there information on the frequency of allelic variants in the population?

The focus of this study was on somatic mutations identified in tumor tissue; the analyzed samples were obtained predominantly from primary lung tumor biopsies (n=108/147; 73.4%).

  1. I would ask the authors to include in the discussion of the results the summary data on all mutations in the EGFR gene, regardless of the type of mutation and exon. It is possible that the total contribution of mutations in EGFR will be comparable to the contribution of mutations in TP53. What mutations were found in TP53?

A paragraph summarizing the total frequency of all EGFR mutations combined (n=28/147; 19.0%) has been added to Results sections to provide a clearer comparison with the prevalence of TP53 mutations (n=61/147; 41.5%): “TP53 mutations were identified in 61 patients, representing the most frequently altered gene in the cohort (41.5%). A wide spectrum of mutations was observed, including missense (52.5%), nonsense (26.2%), frameshift (13.1%), and splice site variants (8.2%), predominantly affecting the DNA-binding domain of the protein. Recurrent hotspot mutations included p.Arg273 (observed as p.Arg273His, p.Arg273Leu, p.Arg273Ser, and p.Arg273Cys, n=5), p.Val157Phe (n=3), p.Cys242Phe (n=2), p.Gly266Val or Ter (n=3), and p.Gly154 (p.Gly154Val or frameshift, n=3). Other pathogenic variants included p.Arg282Trp, p.Arg175His, p.Glu258Lys, p.Arg249Ser, p.Gly245Cys, and p.Arg213Ter. Several splice-disrupting alterations were detected, including c.673-1G>T, c.673-1G>C, c.376-1G>A, and c.920-1G>T, which are predicted to affect mRNA processing. Two patients harbored the synonymous variant p.Thr125=, and a subset showed compound alterations”.

  1. Figure 2. Please increase the font size in the names of genes (mutations). Figures 3-5. The names of genes (mutations) are given in gray. Please change the color of the captions to black. And I would ask the authors to pay attention that all mutations in the EGFR gene in these figures are designated as "EGFR...". This makes it impossible to compare the figure with the text of the article. Please correct it. In addition, it seems to me that there is no caption for the KRAS gene in all three figures.

Figures 2–5 have been updated. Gene names are now in black font and enlarged for readability. KRAS has been explicitly labeled in each figure, and all EGFR variants are consistently labeled. Figures have also been repositioned to follow their first citation in the text as best as possible.